# The Role of Extracellular Vesicles in Atopic Dermatitis

**DOI:** 10.3390/ijms25063255

**Published:** 2024-03-13

**Authors:** Catherine Harvey-Seutcheu, Georgina Hopkins, Lucy C. Fairclough

**Affiliations:** School of Life Sciences, University of Nottingham, Nottingham NG7 2RD, UK; mzych27@exmail.nottingham.ac.uk (C.H.-S.); georgina.hopkins@nottingham.ac.uk (G.H.)

**Keywords:** extracellular vesicles, atopic dermatitis, allergic sensitization, *Staphylococcus aureus*, *Malassezia sympodialis*, mast cells

## Abstract

Atopic dermatitis, or eczema, is the most common chronic skin disorder, characterized by red and pruritic lesions. Its etiology is multifaceted, involving an interplay of factors, such as the allergic immune response, skin barrier dysfunction, and dysbiosis of the skin microbiota. Recent studies have explored the role of extracellular vesicles (EVs), which are lipid bilayer-delimitated particles released by all cells, in atopic dermatitis. Examination of the available literature identified that most studies investigated EVs released by *Staphylococcus aureus*, which were found to impact the skin barrier and promote the release of cytokines that contribute to atopic dermatitis development. In addition, EVs released by the skin fungus, *Malassezia sympodialis*, were found to contain allergens, suggesting a potential contribution to allergic sensitization via the skin. The final major finding was the role of EVs released by mast cells, which were capable of activating various immune cells and attenuating the allergic response. While research in this area is still in its infancy, the studies examined in this review provide encouraging insights into how EVs released from a variety of cells play a role in both contributing to and protecting against atopic dermatitis.

## 1. Introduction

Atopic dermatitis (AD), also known as eczema, the most common chronic skin disorder [1], is a Type 1 Hypersensitivity syndrome, where individuals are predisposed to produce high levels of IgE. This characterizes numerous “atopic” diseases, including allergic asthma, hayfever, and allergic rhinitis [2]. The main characteristics of AD are pruritus, red and exudative lesions, and a fluctuating disease pattern [3]. The pathophysiology of AD is complex, with an interplay of several pathophysiological mechanisms contributing to the etiology and clinical manifestations of AD [1]. These include a dysfunctional epidermal barrier, skin microbe dysbiosis, and T helper 2 cell (Th2)-skewed immune dysregulation [4,5]. The epidermal barrier is weakened in AD due to factors including hereditary mutations in filaggrin, a protein involved in skin barrier formation, and scratching due to itching, which causes mechanical damage [5]. Furthermore, the pH of AD skin is higher than that of normal skin, which affects the skin microbiome, protease function, and mediators of inflammation and pruritus [6]. Damage to the epidermis facilitates the penetration of allergens and their interaction with immune cells. It also induces the release of immunomodulatory alarmin proteins by keratinocytes [1,7].

Common allergens in AD include the bacteria *Staphylococcus aureus* (*S. aureus*) [8] and the fungus *Malassezia sympodialis*, which show dominance in AD skin [7]. These allergens penetrate the damaged skin to trigger the Th2 immune response, whose excessive activation causes AD [5]. The immunopathology of AD, highlighting the role of both immune cells and infection, is presented in Figure 1. Specifically, Th2 activation occurs after dendritic cell (DC) presentation of allergens to naïve T cells, which leads to the secretion of the type 2 cytokines IL-4, IL-5, and IL-13. These induce immunoglobulin E (IgE) antibody class switching in B lymphocytes and the production of antigen-specific IgE. The antibodies enter the circulation and bind to receptors on mast cells and basophils, causing degranulation: the release of histamine, cytokines, and further mediators [2]. In addition to DCs and T cells, macrophages are also involved in the immunopathology of AD, as they are recruited to the diseased tissue and differentiate into various phenotypes. Classically activated macrophages, also called pro-inflammatory M1 cells, are induced by IFN-γ and have a high capacity to present antigens. In contrast, alternatively activated macrophages, also called anti-inflammatory M2 cells, are induced by IL-4, and they promote Th2 responses. Unsurprisingly, there is a significant increase in M2 macrophages in the skin of patients with AD compared with healthy donors, whilst there are no clear changes in M1 infiltration in skin samples of patients with AD [9,10]. 

Extracellular vesicles (EVs) are defined as naturally released cell-derived particles that are delimitated by a lipid bilayer and cannot replicate [11]. Released by all cells, prokaryotes, eukaryotes, and archaea alike, they function as intercellular mediators, delivering biologically active contents to host cells [12,13]. Initially understood as cellular “trash bags”, functioning only to dispose of waste, they have now been identified as key players in both normal physiological functioning and pathological progression [12,14]. Two main subtypes of EVs exist, separated based on their mode of release and size; ectosomes are formed by outward budding of the plasma membrane and sized between 100 and 1000 nm, and exosomes are of endocytic origin and smaller, sized between 30 and 200 nm [13]. EVs contain proteins, lipids, and nucleic acids, but the exact cargo varies according to cell type and physiological conditions; thus, EVs are heterogeneous [15]. The uptake of EVs by recipient cells occurs similarly regardless of EV origin [16]. EVs can be internalized by endocytosis, which is the primary method of internalization [17], but also by phagocytosis [17] or macropinocytosis [18]. Once EVs are internalized by endocytosis, they re-embark on the endosomal recycling pathway, resulting in the delivery of EVs to organelles or the cytoplasm. 

The secretion of EVs can affect antigen presentation, cell differentiation and activation, and immune regulation and suppression. The EVs’ surface markers, cargo, and functions are closely related to the pro-inflammatory or anti-inflammatory properties of the parent cells [19]. Existing research has investigated the roles of EVs in various disease states, from cancer to metabolic and cardiovascular disease [13]. Yet, despite *S. aureus*, *M. sympodialis*, and immune cells involved in AD being capable of releasing EVs, the role of these EVs and their pro- or anti-inflammatory effect on AD is not widely published. Understanding the role of EVs in AD development may be key in determining how to prevent the disease. Of the two types of EVs, the focus of this review is on exosome EVs in AD, as few studies investigate ectosomes in disease. This review aims to highlight the complexity of the interactions giving rise to AD while presenting EVs as key contributors influencing the disease’s pathogenesis. 

## 2. *Staphylococcus aureus*-Derived EVs in Atopic Dermatitis

*S. aureus* is a Gram-positive bacterium that naturally resides on the skin, nasal passages, and respiratory tract as part of the healthy microbiome [20,21]. In AD skin, there is a notable decrease in microbiome diversity, resulting in an overabundance of *S. aureus*. This heightened colonization directly correlates to AD disease severity, as assessed by the scoring atopic dermatitis index [22]. Changes present in AD skin, such as lower anti-microbial peptides, filaggrin, and higher skin pH are thought to allow *S. aureus* to proliferate. *S. aureus* also activates a very large number of T cells abnormally and in an unregulated manner, leading to excessive cytokine release and IgE production by infiltrated immune cells, thus contributing to the immunopathogenesis of AD [23,24]. 

### 2.1. The Secretion of EVs by Staphylococcus aureus 

In 2009, Lee et al. were the first to demonstrate the release of EVs by *S. aureus*. On analyzing thin sections of *S. aureus* using transmission electron microscopy (TEM), they were able to show the formation and secretion of EVs into the extracellular space. The EVs released by *S. aureus* (SEVs) are 20–100 nm in size, spherical, bilayered, and closed membranous structures [25]. The release of EVs from Gram-positive bacteria was initially understood to be impossible due to the thick bacterial cell wall surrounding the plasma membrane. However, the existence of SEVs demonstrates their capacity to somehow traverse it. Indeed, SEVs have been found to contain the surface-associated protein N-acetylmuramoyl-L-alanine amidase, a hydrolase capable of breaking down cell wall glycopeptides [25]. Furthermore, phenol-soluble modulins (PSMs), a family of *S. aureus* virulence factors, were reported to have membrane-damaging activities. It was shown that deletion of the PSM gene resulted in a decrease in the size and quantity of released EVs [26]. It is thought that PSMs increase membrane fluidity, allowing bacterial turgor pressure to drive the release of EVs. 

### 2.2. The Cargo of Staphylococcus aureus-Derived EVs

Lee et al. were also the first to carry out a proteomic analysis of EVs released by *S. aureus*. They identified a total of 90 vesicular proteins, classified into cytoplasmic (56.7%), membrane (16.7%), and extracellular (23.3%) locations. Each protein carries out a specific function. These include metabolism, cell wall biogenesis and organization, bacterial pathogenesis, and response to antibiotics. They identified the presence of key virulence factors in EVs, including superantigens, alpha-hemolysin (causing host cell lysis), coagulase factors, and immunomodulatory proteins like staphylococcal protein A [25]. 

The presence of non-protein components like nucleic acids, lipids, and metabolites in EVs of other Gram-positive bacteria has been well established, but very few studies have investigated the non-protein components of EVs from *S. aureus* specifically [27]. Schlatterer et al. used fluorescent membrane dye to confirm the presence of membranous lipids [28], and Andreoni et al. found that DNA molecules were associated with EVs [29]. However, it was not the primary aim of these studies to investigate EV components, and as such, these have not been thoroughly examined yet. 

### 2.3. The Functions of Staphylococcus aureus EVs in Atopic Dermatitis

The advantages EVs provide the bacterium must be significant enough to outweigh the substantial energy expenditures associated with their manufacture [27]. They transfer proteins, perform cell–cell signaling, eliminate competitors, and deliver toxins to host cells [25]. These abilities help them resist antibiotics, form the biofilm, disrupt the epidermal barrier, and exert immunomodulatory effects in AD [27]. Table 1 summarizes the studies investigating the functions of *S. aureus*-derived EVs in AD.

#### 2.3.1. Antibiotic Resistance

Topical corticosteroids and antibiotic therapies are used to decrease *S. aureus* colonization in AD skin but are not able to eliminate the presence of the bacteria entirely [23]. *S. aureus*-derived membrane vesicles (SEVs) have been found to spread antibiotic resistance to surrounding bacteria; Lee et al. detected that SEVs carry biologically active beta-lactamase (BlaZ), an enzyme capable of inactivating beta-lactam antibiotics like penicillin and cephalosporins. SEVs were able to transfer a transient resistance against ampicillin to surrounding bacteria [30]. This indicates that *S. aureus* uses its EVs to help circumvent the effects of antibiotics, thus evading AD treatment. 

#### 2.3.2. Epidermal Disruption

Keratinocytes are specialized epithelial cells which make up the epidermis of the skin. They recruit immune cells and trigger the Th2 response typical of AD [37]. SEVs can disrupt the keratinocyte barrier, as Jun et al. detected the virulence factor, Staphylococcal protein A (SPA), within SEVs throughout all layers of the epidermis in AD lesions, whereas they were absent in healthy skin [31]. This indicates that *S. aureus* exploits weakness in AD skin to distribute virulence factors contained in SEVs deep into the epidermis which it otherwise cannot access. Indeed, some strains of *S. aureus* produce SEVs which carry alpha-hemolysin, a highly cytotoxic protein. Hong et al. found that those SEVs cause rapid keratinocyte necrosis [32]. The destruction of keratinocytes weakens the epidermal barrier, allowing further pathogenic penetration. Furthermore, when Hong et al. tape-stripped mouse skin and applied different doses of SEVs, they found epidermal thickening in proportion with the amount of SEVs applied and infiltration of the dermis by inflammatory cells [32]. Thickening of the epidermis is also known as lichenification and is a hallmark symptom of AD. 

#### 2.3.3. Immune Response Modulation

Excessive or prolonged inflammation contributes to diseases such as AD [2]. Four studies found that SEVs increase the release of inflammatory cytokines in the skin. Hong et al. found that applying SEVs to mouse dermal fibroblasts in vitro caused an increase in IL-6, thymic stromal lymphopoietin (TSLP), eotaxin, and macrophage inflammatory protein-1alpha (MIP-1alpha) [33]. TSLP and eotaxin have been established to contribute to AD pathogenesis. Expression of TSLP is significantly correlated with itching and skin barrier disruption [38]. It causes the migration and activation of dendritic cells, which in turn activate proliferation and cytokine release from Th2 cells. Eotaxin has also been found to attract and activate Th2 lymphocytes [39]. These release IL-4, IL-5, IL-9, IL-13, and IL-31, causing excessive IgE release and, thus, an AD eruption in the skin [40]. By demonstrating that SEVs can increase TSLP and eotaxin levels, Hong et al. have shown that they directly contribute to the pathogenesis of AD. 

Furthermore, keratinocytes stimulated with SEVs released the pro-inflammatory cytokines IL-1beta, IL-6, IL-8, MIP-1alpha, and TNF-alpha [31,34] which promote redness, swelling, and recruitment of neutrophils and macrophages into the inflamed skin. These cells can worsen inflammation by secreting additional cytokines and recruiting further leukocytes, contributing to the chronic inflammatory state observed in AD [2]. SEV stimulation of HaCaT keratinocytes and dermal fibroblasts also resulted in increased pro-inflammatory cytokine production, specifically IL-6, which inhibits the terminal differentiation of keratinocytes into functional stratum corneum cells [41]. 

SEVs have additionally been found to impact cytokine release in microvascular endothelial cells. Kim et al. found that SEVs could activate endothelial cells, increasing their release of the cell adhesion molecules E-selectin, ICAM-1, and VCAM1, contributing to immune cell infiltration [35]. By increasing the infiltration of immune cells, SEVs contribute to the initiation and maintenance of the inflammatory response.

#### 2.3.4. Biofilm Formation 

*S. aureus* biofilm has been found to contribute to the pathogenesis of AD. Its sheer size and density allow the bacteria to evade phagocytosis by macrophages and neutrophils [42]. Im et al. demonstrated that EVs from *S. aureus* made skin surfaces more hydrophilic. Given that other bacterial species can only adhere to hydrophobic surfaces, while *S. aureus* can form biofilm in both hydrophilic and hydrophobic environments, *S. aureus* uses its EVs to “stake a claim” on surfaces and preferentially form its own biofilm, preventing other pathogens from forming biofilms [36]. This mechanism could be therapeutically beneficial, as it was shown to prevent biofilm formation from a multi-drug-resistant isolate of *Acinetobacter baumanni.* However, it must be noted that *S. aureus* EV biofilm formation did not remove any prior biofilms already formed by other bacteria; thus, its activity is purely preventative [36]. Therefore, resident skin flora, such as *S. epidermidis*, which has been shown to produce EVs that can prevent AD inflammation, would not be affected by the SEV biofilm [43]. The SEV biofilm also exerts a detrimental effect on keratinocytes, triggering apoptosis and the release of TSLP [44]. As such, SEVs indirectly contribute to AD pathogenesis by helping the formation of *S. aureus* biofilm, which shields bacteria from innate immune cells and triggers apoptosis and TSLP secretion from keratinocytes. 

## 3. *Malassezia sympodialis*-Derived EVs in Atopic Dermatitis

*Malassezia* is the most abundant genus of fungi on the skin. It comprises 18 species, but *sympodialis* is associated with AD. As lipophilic yeasts, they tend to colonize areas of high sebaceous gland activity like the face, scalp, breasts, and upper back. The balance between host immunity and fungal activity is a delicate one, requiring enough immune activity from the host to control the number of fungi, but not so much as to cause inflammation [45]. This is referred to as commensalism, which can be disrupted by barrier damage, imiquimod treatment, and allergy, leading to the normally protective antifungal response to become a pathological response [45]. 

EVs from *M. sympodialis* (MalaEx) were first observed in 2011 [46], with sizes ranging from 50 to 600 nm and diverse contents. 

### 3.1. The Secretion of Malassezia sympodialis-Derived EVs 

*Malassezia* fungi have a thick cell wall which provides protection for the cell but is a barrier to the release of EVs. This is highlighted by the fact that fungal mutants with defective cell walls show increased EV release [47]. Like Gram-positive bacteria, it is thought that turgor pressure, or the pressure exerted by the contents of the cell against the cell wall, could play a role in forcing fungal vesicles through the cell wall. Fungal cell walls also contain pores, ranging from 1 to 400 nm in size, which might provide passages for EVs [48]. It has been hypothesized that the hydrolases contained in MalaEx might serve to hydrolyze cell wall components for the passage of EVs [49]. MalaEx have also been found to contain enzymes required for cell wall synthesis. This indicates that the protein components of EVs could help their release by breaking down the cell wall and remodeling it after the fact [50]. 

### 3.2. The Cargo of Malassezia sympodialis-Derived EVs

Like bacterial EVs, fungal EVs are made of a lipid bilayer surrounding a lumen containing proteins, nucleic acids, and carbohydrates. Unlike bacterial EVs, fungal EVs have not been found to be carriers of DNA. However, studies have confirmed the presence of a wide range of RNA in the vesicles, including mRNA, tRNA, and rRNA [51]. Small RNA has been identified in MalaEx specifically, although its function is yet to be elucidated [52]. Proteomic analysis of MalaEx detected a total of 2439 vesicular proteins, the most proteins identified in any fungal EV study [53]. Some of these proteins had hydrolase and catalytic functions. which were found in higher proportion in the vesicles than in the fungal cells [49]. Fungal EV membranes contain ergosterol, a fungi-specific sterol with a role in maintaining membrane integrity, as well as phospholipids [54]. 

### 3.3. The Functions of Malassezia sympodialis-Derived EVs in Atopic Dermatitis

A summary of the studies investigating the functions of *Malassezia sympodialis*-derived EVs in AD is presented in Table 2. These studies show EVs can disseminate allergens, cause skin barrier degradation, or lead to modulation of immune responses.

#### 3.3.1. Dissemination of Allergens

MalaEx contain antigens, or molecules capable of producing an immune response [46]. To determine whether the antigens could initiate an allergic response, serum from an AD patient containing IgE antibodies against *M. sympodialis* was used to search for IgE-binding epitopes on antigens. An IgE-binding protein was detected in the vesicle, confirming the presence of allergens [46]. The vesicles were also enriched in Mala s1 and Mala s5–13 allergens [49]. 

The nanosize of the vesicles is thought to allow their wide dissemination, and thus, the spread of allergens in the skin [49]. This could induce IgE-mediated hypersensitivity reactions on a wider scale, contributing to AD pathogenesis or deterioration. 

#### 3.3.2. Skin Barrier Degradation

MalaEx were found to be enriched with hydrolases and other catalytic proteins. Lysophospholipases specifically are used by the cell to break down sebaceous lipids on the skin, without which *Malassezia* cannot survive. Packaging lysophospholipases in EVs allows for a higher concentration of the enzyme and a more efficient breakdown of sebum [56]. However, the release of irritating unsaturated fatty acids as a byproduct of lipid catalysis causes the degradation of the skin barrier. These have been shown to cause a significant increase in transepidermal water loss, a measure of barrier integrity and irritant skin response [57]. Contributing to barrier degradation allows further penetration of *Malassezia* and *S. aureus* EVs. 

#### 3.3.3. Immunomodulation

Three studies found that MalaEx exerted an immunomodulatory impact relevant to AD. Gehrmann et al. found MalaEx to significantly enhance IL-4 production in peripheral blood mononuclear cells (PBMCs) from AD patients, compared to those from healthy controls, in a dose-dependent manner [46]. IL-4 production can lead to T-cell activation, further Th2 cytokine production, and the perpetuation of AD inflammation, itching, and skin damage. 

Additionally, MalaEx were found to induce a dose-dependent TNF-alpha response in both PBMCs from AD patients and healthy controls [46]. The indifferent reaction between patient groups suggests that the TNF-alpha response occurs regardless of previous sensitization to *M. sympodialis* and is an innate immune response. This is unlikely to be a direct contributor to AD pathogenesis but can increase cutaneous inflammation.

Johansson et al. used confocal laser scanning microscopy to show the internalization of MalaEx by keratinocytes and monocytes, localizing around the keratinocyte nuclei [49]. This uptake was temperature dependent, occurring most prominently at 37 °C, and not at all at 4 °C. Vallhov et al. showed that keratinocytes stimulated with MalaEx exhibited a pro-inflammatory response. Indeed, a 6-fold increase in the expression of ICAM-1 in keratinocytes co-cultured with MalaEx was observed, compared to unstimulated controls [55]. ICAM-1 is an adhesion molecule on epithelial cells with a role in driving the inflammatory response through the recruitment and activation of immune cells. This reaction indicates the capacity of MalaEx to modulate interactions between the skin and the immune system, promoting inflammation.

## 4. Host Mast Cell-Derived EVs in Atopic Dermatitis

Mast cells are the major cell type responsible for atopic reactions [58]. Derived from hematopoietic progenitor cells, they are found in connective tissues under all epithelia [2,59]. They store the major mediators of the atopic reaction including histamine, heparin, proteases, cytokines, and chemokines [60]. Chronic inflammation due to the release of mediators by mast cells occurs when exposure to the allergen is repetitive or continuous, as is typically the case in AD [61]. 

### 4.1. The Secretion of Mast Cell-Derived EVs

Mast cell-derived EVs (MCEVs) are 60–100 nm cup-shaped structures, localized in intracytoplasmic granules [62], suggesting that mast cell granules might be able to release EVs quickly [62,63]. Further studies have shown that mast cells secrete two types of EVs depending on the activation state of the cell [64]. The first type is EVs constitutively released in cells that are not undergoing degranulation (unstimulated MCEVs). The second type is those secreted upon antigen–IgE–FcεRI crosslinking (stimulated MCEVs). Activation of mast cells causes a marked increase in the number of EVs secreted, and these show distinct proteomics and lipidomics compared to constitutively secreted MCEVs [64]. 

### 4.2. The Cargo of Mast Cell-Derived EVs

Unstimulated MCEVs primarily contained CD9 surface proteins, whereas stimulated MCEVs contained high levels of CD63. CD36 is an abundant protein in the membrane of mast cell granules, supporting the hypothesis that stimulated MCEVs might have a granular origin [64]. Additionally, lipidomic analysis showed an enrichment in phosphatidylinositol in stimulated EVs, whilst unstimulated MCEVs were higher in phosphatidic acid. Proteases CPA3 and mMCP4, 5, and 6 were also detected in stimulated MCEVs, suggesting a potential role as carriers of granule mediators [64]. Finally, MCEVs, regardless of cell stimulation status, have been found to contain the antigen-presenting MHC class 2 molecules, CD86, CD40, LFA-1, and ICAM-1 molecules [62]. The presence of these molecules demonstrates that MCEVs possess the necessary molecules to exert an antigen-presenting role. 

### 4.3. The Functions of Mast Cell-Derived EVs

While research is yet to investigate the role of MCEVs in AD specifically, five studies have explored their influence on the T helper 2 immune response, DC activation, and free IgE levels, which characterize atopic diseases (Table 3).

#### 4.3.1. Lymphocyte Activation

Lymphocyte activation is a crucial aspect of the immune response. It occurs when specific antigens are recognized by receptors on the lymphocyte surface, inducing proliferation of the cells and their differentiation into specialized effector lymphocytes. Li et al. investigated the lymphocyte-activating properties of MCEVs [65], in vitro, by co-culture with T cells. They found that when incubated with MCEVs, T cells showed a higher proliferation rate than the control group and demonstrated Th2 differentiation. This would suggest that MCEVs contribute to the pathogenesis of AD by inducing the production of Th2 cytokines. Toyoshima et al. [66] also found that MCEVs can promote a Th2 response. The effects of MCEVs on type 2 innate lymphoid cells (ILC2s) were investigated in vitro. It was found that the co-culture of ILC2s with stimulated MCEVs resulted in a significantly enhanced production of IL-5. However, IL-13 production was unchanged. As such, MCEVs influence the functions of ILC2 cells, promoting the release of type 2 cytokine IL-5, which contributes to AD. Conversely, Skokos et al. [62] found that MCEVs were capable of triggering Th1 responses [69] in vivo. Mice were injected with MCEVs, resulting in cell proliferation and IL-2 and IFN-gamma production. The presence of Th1 cytokines can inhibit Th2 differentiation and suppress Th2 cytokine production. As such, it appears that in vivo, MCEVs might play a protective role in AD by inducing Th1 differentiation. The difference in findings between the in vitro and in vivo studies highlights the importance of further in vivo research, where the experiment is conducted in physiologically relevant conditions. 

#### 4.3.2. Dendritic Cell Maturation and Activation

Dendritic cells are the most effective antigen-presenting cells, capable of stimulating naïve CD4^+^ and CD8^+^ T cells. To exert this function, they must undergo maturation. Skokos et al. determined the ability of MCEVs to induce the maturation of dendritic cells [67], where 48 h of incubation with MCEVs upregulated MHC class 2, CD80, CD86, and CD40. The same experiment was carried out with exosomes from B lymphocytes and macrophages, but these were not found to affect dendritic cell maturation. Additionally, this study found dendritic cells that internalized MCEVs could present the MCEV antigens to induce a significantly more potent activation of T cells than those presented with soluble antigens [67].

#### 4.3.3. Reducing Free IgE

Elevated IgE levels are a hallmark of atopic dermatitis patients, leading to excessive activation of mast cells and basophils, resulting in the release of mediators responsible for the symptoms of allergic disease. Despite the majority of existing research suggesting that EVs play a pro-inflammatory role in AD, Xie et al. demonstrated that MCEVs have the ability to decrease the levels of free IgE [68] and thus promote tolerance instead. Upon incubating MCEVs with IgE for 2 h, a significant reduction in IgE was detected. The study identified the presence of 25–40 FcERI molecules on MCEVs, suggesting their involvement in binding IgE. Furthermore, a significant decrease in mast cell degranulation was noted when mast cells were treated with IgE that had been pre-incubated with MCEVs for 2 h. As such, MCEVs could exert a protective role in atopic diseases including AD, where they reduce free IgE.

## 5. Discussion

Two main contrasting hypotheses have been proposed for the pathogenesis of AD: the “outside-in” hypothesis, where epidermal barrier dysfunction triggers immune activation, and the “inside-out” hypothesis, where AD is primarily cytokine-driven with secondary skin barrier dysfunction [70]. Research into the role of EVs in the pathogenesis of AD suggests that these hypotheses can be integrated, with evidence for AD promotion by the skin pathogen *S. aureus* and the fungus *Malassezia* but also by EVs within the host, primarily mast-cell derived EVs. The varied and important roles of EVs in AD are summarized in Figure 2. 

*S. aureus* is a pivotal contributor to the pathogenesis of AD, by virtue of its adept skin colonization and ability to trigger the immune response. In vitro studies discovered the bacteria’s ability to induce keratinocyte necrosis and increase pro-inflammatory cytokines, some of which are capable of directly fueling AD pathogenesis, while others are known to contribute to a generalized inflammatory response. In vivo studies notably found that SEVs induced increased epidermal thickening, mirroring the characteristics of AD skin. Collectively, these findings hint at a multifaceted contribution of SEVs to furthering the pathogenic endeavors of *S. aureus* within the context of AD. Further in vivo studies using mouse models and physiologically relevant experimental conditions would allow for a deeper understanding of SEV mechanisms in AD skin.

*Malassezia*-derived EVs are the other EV type to have been investigated in the context of AD. This stems from *Malassezia*’s recognized status as an allergen on AD-afflicted skin. MalaEx contain allergens, suggesting a potential involvement in the promotion of allergic responses characteristic of AD. In vitro studies have found that MalaEx induce an increase in pro-inflammatory cytokines, some of which directly contribute to AD pathogenesis. In vivo studies on MalaEx are yet to be conducted. 

The final focus of existing literature investigating EVs in AD was EVs released by mast cells (MCEVs). Whilst this literature is not specific to the skin, it is still relevant to AD. Functionally, MCEVs play a concrete role in immune interactions, inducing dendritic cell maturation, reducing free IgE, and inducing lymphocyte proliferation and cytokine production. Thus, the literature investigating MCEVs shows conflicting results, with evidence for enhancing AD pathogenesis but also for a protective role. 

As well as involvement in the pathogenesis of AD, EVs have also been proposed as therapeutic agents for the treatment of AD. Of the limited research available, exosomes from human adipose tissue-derived mesenchymal cells were found to effectively treat symptoms of AD, resulting in a reduction in clinical score, IgE levels, eosinophils and mast cells, and inflammatory cytokine expression [71]. A major finding is the evidence for EVs in wound healing and tissue regeneration by mechanisms such as revascularization, cell proliferation, motility, and neo-angiogenesis [72]. Furthermore, EVs have largely been discussed as biomarkers for cancer, but some research also identifies EVs as potential biomarkers in AD. For instance, serum microbial EVs can be used as biomarkers in AD due to large differences in microbial diversity in AD patients and healthy controls [73]. Importantly, EVs have also been shown to act as drug delivery vehicles due to their structure and stability in circulation. One example is the delivery of melatonin, which is an effective treatment of AD, but its transdermal delivery is a challenge. Thus, one study utilized melatonin-loaded extracellular vesicle-mimetic nanoparticles for efficient transdermal delivery of melatonin, which resulted in improved AD symptoms [74]. EVs from *Staphylococcus* epidermidis have also been highlighted as potential bioactive nanocarriers for transdermal drugs, with evidence for increased skin penetration and cellular uptake, compared to delivery without EV encapsulation [43].

In terms of future work, more in vivo research is required. The clear lack of in vivo investigations causes an impediment to the progression of research on EVs in AD. These are preferable to in vitro studies due to the multifactorial nature of AD and the complex interactions involved. Studies of living organisms present a more realistic picture of the biological environment than isolated cell cultures. However, they are more difficult to undertake due to challenges in EV visualization and tracking in living organisms. While the in vitro studies presented on *S. aureus* and *Malassezia* provide a foundational exploration that offers some preliminary insight into their potential contributions to AD, in vivo studies are necessary to push the inquiry into clinically relevant territory and potentially provide a therapeutically relevant understanding. 

Aside from a lack of in vivo studies, the main limitation of research in this area is the methodological heterogeneity exhibited by the various studies. There were variations in the concentration of SEVs used in the different studies, as well as the duration of co-incubation of SEVs with cells. Thus, future research involving EVs in AD needs more established methodologies to follow.

Another area lacking was research on EVs influencing macrophage polarization in AD. During a bacterial infection, immune cells like macrophages can uptake bacterial EVs through different TLR pathways or endocytosis and subsequently release pro-inflammatory or anti-inflammatory cytokines, which can in turn affect the functions of other macrophages nearby [75]. However, there is currently no evidence on whether EVs produced by S. aureus in AD promote M1 or M2 macrophages at the site of inflammation. Likewise, evidence for the role of fungus-derived EVs in driving macrophage differentiation in AD is lacking, with studies in other disease settings [76] highlighting that this could be a potential mechanism in AD. Further research is needed to address these important questions.

Overall, this comprehensive exploration of studies on EVs in AD provides an overview of their potential contributions to AD pathogenesis and underscores the need to advance research through in vivo studies, using mouse models and physiologically relevant conditions to pave the way for a deeper understanding and therapeutically relevant insights.

## Figures and Tables

**Figure 1 ijms-25-03255-f001:**
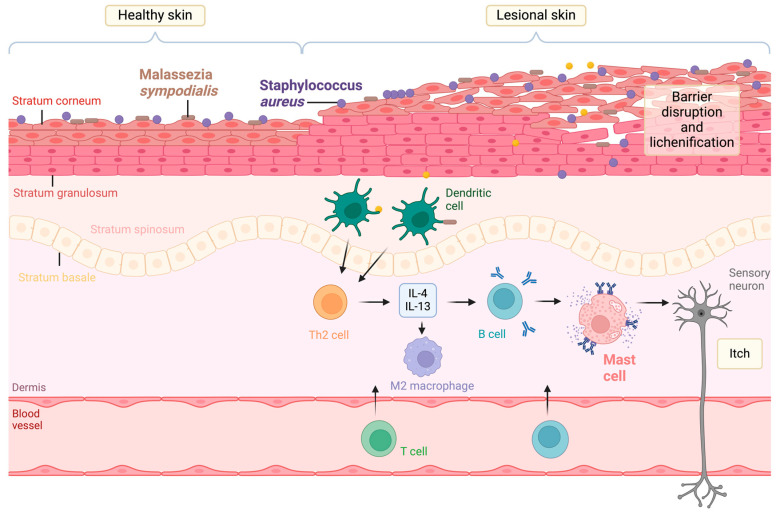
Immunopathology of AD: a comparison of healthy and lesional skin. *S. aureus* and *Malassezia* are present on healthy skin but overabundant on lesional skin. The damaged skin barrier facilitates their entry. They act as allergens on AD skin, triggering the allergic response through their presentation by dendritic cells to T cells. These differentiate into T helper 2 cells and secrete IL-4 and IL-13 cytokines, which induce the secretion of IgE by B cells and also promote M2 macrophage differentiation. This IgE binds to receptors on mast cells and causes degranulation, which is the release of mediators like histamine. These activate sensory nerves, causing itchiness. Scratching follows, which further damages the skin barrier, perpetuating the cycle. Created using BioRender.

**Figure 2 ijms-25-03255-f002:**
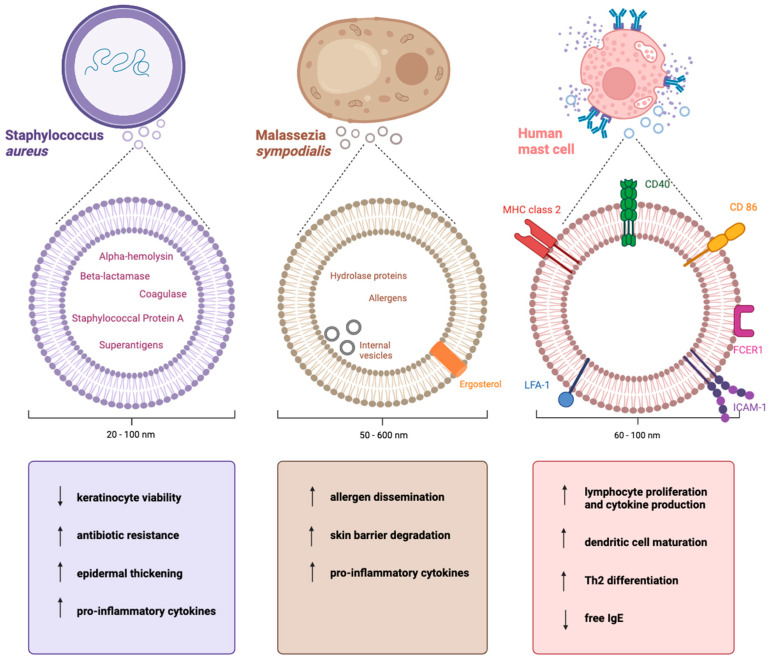
Summary of cells of origin, characteristics, and effects of *S. aureus*-, *M. sympodialis*-, and mast cell-derived EVs. SEVs decrease keratinocyte viability and increase antibiotic resistance, epidermal thickening, and the release of pro-inflammatory cytokines. The direction of effect seems to be towards a promotion of AD pathogenesis. Similarly, MalaEx increase allergen dissemination, skin barrier degradation, and the release of pro-inflammatory cytokines. However, by increasing lymphocyte activation, dendritic cell maturation, and Th2 differentiation while decreasing free IgE, MCEVs seem to have a protective tendency in AD. Created using BioRender.

**Table 1 ijms-25-03255-t001:** Summary of studies investigating the functions of *Staphylococcus aureus* EVs in AD.

Author, Year [Ref]	Model System	Conclusion
Lee et al., 2013 [30]	Bacterial, in vitro	SEVs mediated the survival of BlaZ
Jun et al., 2017 [31]	Human, in vitro	SEVs detected on surface and in cytoplasm of keratinocytes as well as in intercellular space of epidermis. HaCaT cells increased expression of IL-1beta, IL-6, IL-8 and MIP-1alpha when treated with SEVs
Hong et al., 2014 [32]	Human, in vitro	HaCaT viability significantly decreased upon treatment with SEVs from AD patients
Hong et al., 2011 [33]	Murine, in vitro Murine, in vivo	Dermal fibroblasts increased production of IL-6, TSLP, MIP-1alpha, and eotaxin when treated with SEVs. SEV application in vivo caused epidermal thickening and infiltration of dermis by mast cells and eosinophils
Staudenmaier et al., 2022 [34]	Human, in vitro	SEVs induced CXCL8 and TNF-alpha expression in keratinocytes.
Kim et al., 2019 [35]	Human, in vitro	SEVs increased expression of E-selectin, VCAM1, ICAM-1, and IL-6 in HDMECs.
Im et al., 2017 [36]	Bacterial, in vitro	SEVs dose-dependently inhibited *A. baumannii* biofilm development

**Table 2 ijms-25-03255-t002:** Summary of studies on the functions of *Malassezia sympodialis* EVs in AD.

Author, Year [Ref]	Methods	Conclusion
Gehrmann et al., 2011 [46]	Human, in vitro	MalaEx significantly enhanced IL-4 and TNF-alpha production in CD14- and CD34-depleted PBMCs
Johansson et al., 2018 [49]	Human, in vitro	MalaEx contain Mala s1 and s5–13 allergens. MalaEx are internalized by keratinocytes and monocytes.
Valhov et al., 2020 [55]	Human, in vitro	MalaEx induce a dose-dependent increase in ICAM-1 expression in keratinocytes.

**Table 3 ijms-25-03255-t003:** Summary of studies investigating the functions of mast cell EVs in atopic dermatitis.

Author, Year [Ref]	Methods	Conclusion
Skokos et al., 2001 [62]	Murine, in vivo	Proliferation of spleen and lymph node cells as well as IL-2 and IFN-gamma production. Bone-derived mast cells require pretreatment with IL-4 to secrete MCEVs.
Li et al., 2016 [65]	Murine, in vitro	MCEVs induced a higher proliferation rate of T cells than control and a higher proportion of Th2 differentiated cells.
Toyoshima et al., 2021 [66]	Human, in vitro.	MCEVs enhanced IL-5 production in ILC2s in the presence of IL-33. They did not enhance IL-13 production.
Skokos et al., 2003 [67]	Murine, in vivo	MCEVs induced the upregulation of DC maturation markers MHC II, CD80, CD86, and CD40.
Xie et al., 2018 [68]	Murine, in vitro	Each MCEV contained 25 to 40 FcERI molecules. MCEVs bind to free IgE via FcERI, reducing mast cell activation.

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
