# Peer review of "The Role of Extracellular Vesicles in Atopic Dermatitis"

_ijms, 2024, doi:10.3390/ijms25063255_

Round 1

Reviewer 1 Report

Comments and Suggestions for Authors

The manuscript submitted by Harvey-Seutcheu et al is a nice piece of work about the role of extracellular vesicles in the promotion and attenuation of atopic dermatitis. The speech is clear, and the manuscript is concise and highly readable and understandable. However, the soundness and impact of the work could be improved including the following points:

1) The role of S.aureus EVs in atopic dermatitis is well explained. Specifically the antibiotic resistance development, epidermal disruption process and biofilm formation. However, regarding the immune response modulation only in vitro studies involving fibroblasts, keratinocytes and endothelial cells were discussed. Nowadays, macrophage polarization to M1 (pro-inflammatory) and M2 (anti-inflammatory) phenotypes is a hot-topic and a growing field of research. It has been observed that bacteria-EVs are involved in these polarization processes. The manuscript would be enriched if this dichotomy, since the role of EVs on macrophage polarization is not totally elucidated, is included. Do SEVs promote the M1 or M2 polarization? Could SEVs be used for AD attenuation via M2 polarization? Are in vivo (and/or in vitro) studies using bacterial-EVs or SEVs reported in literature for the treatment of inflammatory skin conditions?

2) Same considerations as 1) for MalaEx or other fungi-EVs.

3) The use of EVs as drug delivery systems and carriers is currently widely explored. They have been used as a platform to encapsulate active ingredients (from molecules to siRNAs) due to their capability to act as biomimetic entities is used to target specific cell types. The manuscript would be benefited if it includes a section about the therapeutical use of EVs for treating skin inflammatory disorders (including bacterial-, plant- and cell-derived; natural/bare or loaded with APIs). Are some of these types of formulations commerzialised or under medical/clinical trials? A summarised table of the clinical trials is highly recommended. 

4) Future prospects section is needed to offer an idea about the potential therapeutic and cosmetic applications of the EVs knowledge in the treatment of AD. Could EVs be applied to other immuno-inflammatory skin conditions, such as psoriasis?

Author Response

Thank you for your helpful suggestions, these have now been addressed in the revised manuscript and details are provided below.

1) The role of S.aureus EVs in atopic dermatitis is well explained. Specifically the antibiotic resistance development, epidermal disruption process and biofilm formation. However, regarding the immune response modulation only in vitro studies involving fibroblasts, keratinocytes and endothelial cells were discussed. Nowadays, macrophage polarization to M1 (pro-inflammatory) and M2 (anti-inflammatory) phenotypes is a hot-topic and a growing field of research. It has been observed that bacteria-EVs are involved in these polarization processes. The manuscript would be enriched if this dichotomy, since the role of EVs on macrophage polarization is not totally elucidated, is included. Do SEVs promote the M1 or M2 polarization? Could SEVs be used for AD attenuation via M2 polarization? Are in vivo (and/or in vitro) studies using bacterial-EVs or SEVs reported in literature for the treatment of inflammatory skin conditions? 2) Same considerations as for MalaEx or other fungi-EVs.

Thank you for bringing this to our attention. We agree macrophage polarization in atopic dermatitis is key, so we have now introduced this in lines 48-56. However, we did not find any literature investigating the role of bacterial or fungi EVs influencing M1/M2 polarization in atopic dermatitis. So, we have discussed this lack of literature in lines 439-447.

3) The use of EVs as drug delivery systems and carriers is currently widely explored. They have been used as a platform to encapsulate active ingredients (from molecules to siRNAs) due to their capability to act as biomimetic entities is used to target specific cell types. The manuscript would be benefited if it includes a section about the therapeutical use of EVs for treating skin inflammatory disorders (including bacterial-, plant- and cell-derived; natural/bare or loaded with APIs). Are some of these types of formulations commerzialised or under medical/clinical trials? A summarised table of the clinical trials is highly recommended. 

We agree the manuscript would benefit from a therapeutics section, so we have now included a section on EVs in the treatment and diagnostics of AD in the discussion (lines 406-423).

4) Future prospects section is needed to offer an idea about the potential therapeutic and cosmetic applications of the EVs knowledge in the treatment of AD. Could EVs be applied to other immuno-inflammatory skin conditions, such as psoriasis?

Thank you for this suggestion. In addition to the therapeutics section added in the discussion, we have followed this with some key future perspectives regarding EV work in atopic dermatitis.

Reviewer 2 Report

Comments and Suggestions for Authors

Please see attached PDF for comments.

Comments on the Quality of English Language

English seems fine. Somewhat simplistic but OK.

Author Response

Thank you for your helpful suggestions, these have now been addressed in the revised manuscript and details are provided below.

Introduction

  1. The Introduction needs to first explain atopic dermatitis and links to other pathologies, such as asthma, so that the role of vesicles as a linker between the two diseases can be fully delineated. Suggest moving the second paragraph up and also the figure with it. The current first paragraph can be moved. Also, please talk about the pH differences seen between AD and normal skin. This is an important mediator of disease severity. The Intro should explain AD, but the existing text can be moved into Section 2.0. Extracellular vesicles. Obviously, the other sections should be moved down as well.

Thank you for this suggestion. We have now re-arranged the introduction, and discussed AD in more detail, including the importance of pH on lines 34-36. We felt it was better to keep the AD and EVs section together in the introduction, but we have now discussed AD first, moving the EV section towards the end of the introduction.

  1. You MUST discuss, at least briefly, the immune cell types involved in AD, especially the role of antigen presenting cells and the failure of immune regulatory mechanisms. The DCs mentioned in Figure 1 are not mentioned in the actual text with enough detail to understand the molecular pathway of errant activation and autoimmune propagation. The complete T cell mechanism of autoimmunity must also be strengthened.

Thank you for bringing this to our attention. We have now included a section on the immune mechanisms behind AD and the different cells involved in lines 43-56.

  1. A small note, but proper scientific names are always italicized and given as an italicized capital letter with a period after when used as an abbreviation, e.g., S. aureus, not S aureus.

Thank you for pointing this out, we have now corrected all scientific names in the manuscript.

  1. You may also want to give a short treatise on the construction of EVs, especially with regard to similarities between disease-mediating EVs released by human cells and disease-exacerbating or -causing EVs released by bacteria. There is obviously an immune component to this, so please detail possible mechanisms for EVs released to interact with the immune system.

We agree with this suggestion and an overview of the construction of EVs and their possible pro- and anti-inflammatory roles are mentioned in lines 73-90.

Section 2.0

General Comment: The tables have a lot of unnecessary information. Suggest removing the “Aim” column and condensing the Methods column to a “Model System” to indicate in vivo, in vitro, murine, etc. The conclusions can also be shortened to specific genes or exact descriptions, e.g., “BlaZ mediates survival” instead of “SEVs carry BlaZ, a beta-lactamase protein SEVs mediated the survival of ampicillin susceptible gram negative and positive acteria in the presence of ampicillin.”

Thank you for this suggestion, we have taken your advice and the tables do read much nicer.

  1. What about tolerance mechanisms? Are they affected by EVs? Is the lichenification effective in preventing hemolysis of the keratinocyte barrier layer? Does immunosuppression work to reduce the AD flareups?

Thank you for bringing this to our attention. There is evidence for EVs also influencing tolerance mechanisms, however, of the available literature presented here, the results indicate Th2 responses are promoted, and there is little evidence of tolerance promotion. However, in section 4.3.3, we discuss one study which suggests mast cell-derived EVs reduce free IgE, suggesting there is some evidence for EVs promoting tolerance.  

  1. Detailing the effect of SEVs on crowding out other pathogens is crucial here. Do resident flora also create competitive biofilms on human skin? Does pH matter? Previous evidence is that beneficial skin flora are important in mediating inflammation: DOI: 10.1016/j.jid.2023.02.023.

We agree this is crucial to include. So we have now discussed this in lines 199-208, including the reference provided, thank you.

Section 3.0

  1. The importance of immune regulation of M. sympodialis is mentioned in the first paragraph but not elucidated in Section 3.3. What is known or suspected to cause a breakdown in the balance between fungal count and immune control?

Thank you for pointing this out. We have now elucidated this in lines 215-220.

Section 4.0

  1. Section 4.3.3: This contradicts the previous evidence that MCEVs are activators of disease. This reduction in free IgE is more likely to be from compensatory mechanisms or feedback inhibition or regulatory T cells, etc. Please expand this section to fully outline the mechanisms of IgE modulation.

Thank you for your point. This study does show conflicting evidence compared to the other studies presented in this review, but the results do clearly suggest mast cell-derived EVs reduce free IgE, suggesting there is some evidence for EVs promoting tolerance. 

Discussion

  1. It is strongly recommended to discuss the two competing hypotheses of AD pathogenesis as outline here: DOI: 10.1038/s41423-023-00992-4. The section as written may inflate the role of the skin microbiome without direct clinical evidence that it contributes the largest share of disease pathogenesis and exacerbation.

Thank you for your suggestion. We have now introduced the two competing hypotheses at the beginning of the discussion and described how the current literature suggests the two hypotheses can be integrated when reviewing literature regarding EVs in AD.

  1. The Discussion as written does not synthesize the information presented effectively and simply states what is found in the literature. If this section is to be retained, please discuss the information in the context of the two hypotheses (contrast and compare) and offer at least 3 or 4 SPECIFIC future directions needed to answer the question as to which is correct. If a treatment review section is added, then that topic would be excellent for this section, since clinical papers are always welcome to provide important context to the basic translational studies (i.e., how would existing treatments or new delivery systems exploiting artificially made nanovesicles be useful in ablating the AD response from these exogenous EVs?)

We agree the manuscript would benefit from a therapeutics section, so we have now included a section on EVs in the treatment and diagnostics of AD in the discussion (lines 406-423).

Round 2

Reviewer 2 Report

Comments and Suggestions for Authors

The authors are to be commended for accomplishing all suggested edits.

Comments on the Quality of English Language

There are still some issues (ex: S aureus instead of S. aureus) but overall language is OK.